# Genome Editing Using Cas9 Ribonucleoprotein Is Effective for Introducing *PDGFRA* Variant in Cultured Human Glioblastoma Cell Lines

**DOI:** 10.3390/ijms24010500

**Published:** 2022-12-28

**Authors:** Taiji Hamada, Seiya Yokoyama, Toshiaki Akahane, Kei Matsuo, Akihide Tanimoto

**Affiliations:** Department of Pathology, Kagoshima University Graduate School of Medical and Dental Sciences, 8-35-1 Sakuragaoka, Kagoshima 890-8544, Japan

**Keywords:** cultured glioblastoma cell lines, single-nucleotide substitution, Cas9 ribonucleoprotein, transfection methods, *PDGFRA*

## Abstract

Many variants of uncertain significance (VUS) have been detected in clinical cancer cases using next-generation sequencing-based cancer gene panel analysis. One strategy for the elucidation of VUS is the functional analysis of cultured cancer cell lines that harbor targeted gene variants using genome editing. Genome editing is a powerful tool for creating desired gene alterations in cultured cancer cell lines. However, the efficiency of genome editing varies substantially among cell lines of interest. We performed comparative studies to determine the optimal editing conditions for the introduction of platelet-derived growth factor receptor alpha (*PDGFRA*) variants in human glioblastoma multiforme (GBM) cell lines. After monitoring the copy numbers of *PDGFRA* and the expression level of the PDGFRα protein, four GBM cell lines (U-251 MG, KNS-42, SF126, and YKG-1 cells) were selected for the study. To compare the editing efficiency in these GBM cell lines, the modes of clustered regularly interspaced short palindromic repeat (CRISPR)-associated protein 9 (Cas9) delivery (plasmid vs. ribonucleoprotein (RNP)), methods of transfection (lipofection vs. electroporation), and usefulness of cell sorting were then evaluated. Herein, we demonstrated that electroporation-mediated transfer of Cas9 with single-guide RNA (Cas9 RNP complex) could sufficiently edit a target nucleotide substitution, irrespective of cell sorting. As the Cas9 RNP complex method showed a higher editing efficiency than the Cas9 plasmid lipofection method, it was the optimal method for single-nucleotide editing in human GBM cell lines under our experimental conditions.

## 1. Introduction

In precision medicine, variants in oncogenes and cancer-suppressor genes can be analyzed using next-generation sequencing (NGS), and appropriate molecular targeted drugs can be more effectively chosen for cancer treatment based on the results of NGS analysis [1,2]. However, many variants of uncertain significance (VUS) can be simultaneously detected using NGS with unclear oncogenic functions [3,4]. One approach for the elucidation of VUS is a functional study using cultured cancer cell lines that harbor gene variants of interest generated by genome editing. Genome editing technology, which is used to substitute target sequences with desired mutations, is an effective tool for research in cancer genomics [5,6,7].

VUS is introduced to cultured cells using genome editing techniques, including the clustered regularly interspaced short palindromic repeat (CRISPR)/CRISPR-associated protein 9 (Cas9) system with donor DNA [8,9]. We previously generated human embryonic kidney (HEK) 293T cells harboring a VUS (c.1403A>G) of the platelet-derived growth factor receptor alpha (*PDGFRA*) gene using the CRISPR/Cas9 system, which has been identified as potentially pathogenic [10]. To further examine this variant, we aimed to generate human glioblastoma multiforme (GBM) cells harboring this variant, as *PDGFRA* is one of the most crucial driver genes in GBM [11,12]. Although genome editing studies in cultured human GBM cells have been reported, most of these reports involved the generation of gene knockout cells [13]. To the best of our knowledge, studies regarding genome editing for single-nucleotide substitutions are rarely reported in human GBM cell lines, except for one study that introduced *TERT* promoter mutations using a base editing technique in GBM cell lines [14].

In the present study, we evaluated whether single-nucleotide substitution genome editing could be performed in human GBM cell lines. As efficient Cas9 and single-guide RNA transfers are crucial for successful genome editing, Cas9 with single-guide RNA complex (Cas9 ribonucleoprotein (RNP) complex)-based editing and established Cas9 expression plasmid-based editing were employed to estimate efficient Cas9 delivery in GBM cells. Furthermore, the utility of cell sorting and Cas9 expression plasmid modification were evaluated. The specific aim of the present study was to determine the optimal conditions for single-nucleotide editing of *PDGFRA* in cultured human GBM cell lines.

## 2. Results

### 2.1. Copy Number of PDGFRA and Expression of PDGFR in GBM Cell Lines

Cell polyploidy contributes to genome editing efficiency and consequent cellular phenotypic changes [15]. Therefore, fluorescence in situ hybridization (FISH) analysis of *PDGFRA* was performed to confirm the ploidy of GBM cell lines. A172, NMC-G1, and KNS-42 cells harbored two or three copies of *PDFGRA*, and SF126 and YKG-1 cells showed two copies of *PDGFRA*. In contrast, U-251 MG and T98G cells harbored higher *PDGFRA* copy numbers: 6.0 and 3.67, respectively. The signal ratio of *PDGFRA*/chromosome enumeration probe for chromosome 4 (CEP4) in the GBM cell lines was 1.00, except for U-251 MG cells, in which the *PDGFRA*/CEP4 signal ratio was more than 2.0 (Figure 1A and Table 1).

Next, the expression levels of PDGFRα and PDGFRβ, translated from *PDGFRA* and *PDGFRB*, respectively, were examined in GBM cell lines (Figure 1B). Higher PDGFRα expression levels were observed in U-251 MG (high copy number of *PDGFRA*) and KNS-42 (near-normal copy number of *PDGFRA*) cells, indicating that the copy number of *PDGFRA* did not correspond to PDGFRα expression levels. The expression level of PDGFRβ, an isoform of PDGFR, was lower in KNS-42 cells than in other cell lines; thus, these cells would be less susceptible to the effects associated with PDGFRβ-mediated signaling. The expression levels of PDGFRα well corresponded well to those of *PDGFRA* messenger RNA in the GBM cell lines (Figure 1C).

### 2.2. Comparison of Transfection Efficiency and Cleavage Activity between Cas9 Expression Plasmid Lipofection and Cas9 RNP Complex Electroporation in GBM Cell Lines

Based on the results from the ploidy and Western blot analyses, the following studies were performed in the four GBM cell lines: U-251 MG and KNS-42 cells showing higher PDGFRα expression and SF126 and YKG-1 cells showing a normal copy number of *PDGFRA*. To evaluate the transfection efficiency of the Cas9 expression plasmid, a plasmid that simultaneously expressed Cas9 nuclease and mNeonGreen (pX330AGmt-1×2, truncated chicken β-actin hybrid (CBh)-driven, 9.8 kbp) was constructed (Appendix A). The coding sequence of *Cas9* derived from *Streptococcus pyogenes* is approximately 4.1 kbp. To assess the effect of plasmid size on transfection efficiency, nearly equal molar amounts of a green fluorescent protein (GFP) expression plasmid (4.7 kbp) were also transfected in each experiment. The transfection efficiency of the Cas9 expression plasmid was compared with that of the GFP expression plasmid.

The transfection efficiency of the GFP plasmid using lipofection was higher than that of the Cas9 expression plasmid in all GBM cell lines (Figure 2A and Appendix A), indicating that a larger Cas9 expression plasmid is unfavorable for transfer into GBM cells [16,17]. In the T7 endonuclease I (T7E1) assay, cleavage activity was not detected in the Cas9 plasmid-mediated genome editing. However, genome editing using electroporation of the Cas9 RNP complex showed higher cleavage activity in all GBM cells (Figure 2B and Appendix A).

To investigate the variation in cleavage activity, Cas9 protein expression levels were examined using Western blot analysis. The Cas9 protein expression was induced by electroporation of the Cas9 RNP complex into all GBM cells (Figure 2C). YKG-1 cells expressed lower levels of Cas9 than other GBM cell lines using the Cas9 expression plasmid. This lower level of Cas9 expression could not be attributed to cleavage activity (Figure 2B).

In other cancer cell lines (HEK293T, HCT 116 colorectal cancer, BxPC-3 pancreatic cancer, and HEC50B endometrial cancer cell lines), the transfection efficiency of the Cas9 expression plasmid using lipofection was lower than that of the GFP expression plasmid. Cleavage activity after electroporation of the Cas9 RNP complex was higher than that after Cas9 expression plasmid lipofection. These results were in accordance with those obtained for GBM cell lines (Appendix A).

### 2.3. Single-Nucleotide Editing Using Cas9 RNP Complex and Single-Strand Oligodeoxynucleotide (ssODN)

The cleavage activity induced by Cas9 expression plasmid lipofection was very low, and it may be insufficient to edit *PDGFRA* by plasmid-mediated Cas9 expression. Genome editing using Cas9 RNP complex electroporation was suggested to be optimal in the GBM cell lines, and genome editing for *PDGFRA* was conducted using the Cas9 RNP complex and ssODN (as donor DNA) in the four GBM cell lines (Figure 3A). The ssODNs were prepared from an asymmetric protospacer adjacent motif (PAM) strand ssDNA donor (as the strand containing the NGG PAM sequence), according to a previous study [18] (Figure 3B).

The editing efficiency (number of reads with the desired edit/number of total reads) was measured using NGS analysis (Figure 3C). Targeted editing reads were detected in all GBM cell lines, and the editing rates were 3.6–6.3% (U-251 MG: 4.9%, KNS-42: 3.6%, SF126: 6.3%, YKG-1: 4.2%; without cell sorting), suggesting that the Cas9 RNP complex electroporation method would be sufficient for single-nucleotide editing in GBM cells. The editing rate in *PDGFRA*-amplified U-251 MG cells was similar to that in other GBM cells harboring a near-normal copy number of *PDGFRA*.

To improve editing efficiency, Cas9 RNP complex-electroporated cells were isolated using a cell sorter with fluorescence-labeled tracrRNA (Figure 3A and Appendix A). The edited KNS-42 cells were enriched by cell sorting, and the proportion of edited cells increased nearly two-fold after cell sorting. However, the cell sorting procedure did not enhance the enrichment of edited cells in other GBM cell lines (Figure 3C).

### 2.4. Promoter Change Has Little Effect on Cas9 Expression and Cleavage Activity in GBM Cell Lines

Well-reported editing techniques such as the generation of knockout cells, base editing [19], and prime editing [20] are plasmid-based editing methods. Thus, we hypothesized that plasmid-based editing in GBM cells may be improved by increasing cleavage activity using other promoter-driven Cas9 expression plasmids. To evaluate the effect of promoters on Cas9 expression, Cas9 and mNeonGreen expression plasmids with two different promoters (cytomegalovirus (CMV) and elongation factor 1α (EF-1α)) were constructed. The Cas9 expression level was then estimated by counting the number of mNeonGreen-positive cells.

The number of mNeonGreen-positive cells increased slightly when plasmids with the CMV promoter were used in comparison with the CBh promoter in KNS-42 and YKG-1 cells (Figure 4A). However, the T7E1 assay revealed a slight change in cleavage activity after promoter exchange in KNS-42 and YKG-1 cells (Figure 4B and Appendix A). In HEK293T cells, transfection efficiency and cleavage activity improved with the EF-1α-driven Cas9 expression plasmid (Figure 4A,B).

### 2.5. Genome Editing Using Cas9 Plasmid and Electroporation in GBM Cells

To improve the transfection efficiency, a Cas9 expression plasmid with a CMV promoter was electroporated into GBM cells. Transfection efficiency using electroporation was higher in both Cas9 and GFP expression plasmids compared with using lipofection (Figure 5A).

However, after electroporation using the plasmid-based method, editing was not successfully processed as the targeted *PDGFRA* variant was rarely detected in U-251 MG, SF126, and YKG-1 cells (Figure 5B). The targeted *PDGFRA* variant was only detected in KNS-42 cells, in which editing efficiency significantly increased after cell sorting. At present, we cannot explain the reason discrepancy is observed in transfection and editing efficiency, although it is easy to say that it would have resulted from the different characteristics of the GBM cell line used in this study.

### 2.6. Comparative Analysis of Gene Modification Rate between the Cas9 RNP Complex- and Cas9 Plasmid-Mediated Editing in GBM Cells Using Electroporation

Since the lower editing efficiency in plasmid-based editing was speculated to be due to the lower occurrence of double-strand breaks (DSB), the gene modification rate (number of reads with modifications (insertion, deletion, and substitution)/number of total reads) that reflects the frequency of DSB was analyzed using NGS in Cas9 expression plasmid- and Cas9 RNP complex-mediated editing.

In studies using the Cas9 expression plasmid, the rate of gene modification increased after cell sorting in all GBM cell lines. A higher modification rate was observed in KNS-42 cells after sorting (Figure 6A). However, these modification rates from Cas9 expression plasmid-mediated editing did not reach those of Cas9 RNP complex-mediated editing, even after cell sorting (Figure 6B). When editing using the Cas9 RNP complex, a higher modification rate was achieved with and without cell sorting in all GBM cell lines.

## 3. Discussion

Genome editing is an essential technique used in molecular biology to insert, delete, and substitute nucleotides for evaluating gene function and mutational analysis in cancer genomics [9,21,22]. Our results suggest that genome editing using the Cas9 RNP complex was effective in introducing a *PDGFRA* variant into GBM cell lines. This result is in accordance with previous studies suggesting that Cas9 protein transfection enhances genome editing in mammalian and human cells [23,24]. In our laboratory, the optimal method for single-nucleotide editing of *PDGFRA* in cultured human GBM cell lines is to use the Cas9 RNP complex, ssODN as the donor DNA, and electroporation for transfection, which would be much more convenient than the method using viral vector-mediated Cas9 transfer and cell sorting. In this experimental condition, the off-target effect using the Cas9 RNP complex is expected to be lower than that using the Cas9 expression plasmid [23].

Transfection efficiency is an important factor for regulating efficient genome editing and is affected by the delivery method of plasmids, oligonucleotides, and RNPs [23]. In the present study, we showed that electroporation increased plasmid-mediated transfection efficiency; however, this did not contribute to a subsequent increase in editing efficiency. In contrast, the combination of the Cas9 RNP complex and electroporation resulted in higher transfection efficiency, Cas9 expression, and editing efficiency. Thus, editing experimental conditions should be evaluated based on the final editing efficiency of the genes of interest, not merely by transfection efficiency. In addition to lipofection and electroporation, viral vector-mediated gene transfer is an alternative method to improve transfection efficiency [25]. In recent studies, recombinant adeno-associated virus donor vector-mediated gene transfer was effective for Cas9 delivery in homology-directed genome editing [26]. However, the packaging capacity of viral vectors is limited, and it is difficult to produce viral vectors carrying engineered Cas9 because of their large gene size [27,28]. Furthermore, the *Cas9* gene might integrate into unexpected site(s) after lentivirus and retrovirus vector-mediated gene transfer, which would cause undesired effects on cell functions; therefore, genome editing using viral vectors should be performed with caution [29].

The editing efficiency was lower in Cas9 expression plasmid-mediated editing in GBM cells, except for KNS-42 cells, than in Cas9 RNP complex-mediated editing. Although the editing efficiency and DSB frequency improved after the cell sorting procedure to some degree, this increase was not sufficient for plasmid-mediated single-nucleotide gene editing. As single-nucleotide editing in the present study was mediated through a homologous direct repair pathway, higher cleavage activity is necessary for genome editing of nucleotide substitutions [30,31]. In contrast, under our experimental conditions, indel mutations were generated using Cas9 expression plasmid transfer. Therefore, the increased DSB frequency following cell sorting may be sufficient to generate a knockout cell, resulting in a frameshift mutation mediated through a non-homologous end-joining-mediated repair pathway [32,33]. As authorized editing techniques such as the generation of knockout cells, base editing, and prime editing are mostly plasmid-mediated techniques [19,20], our present results do not conclude the uselessness of plasmid-based editing for the production of knockout cells and the introduction of other gene mutations.

Chromosomal polyploidy and genetic copy number alterations (gene loss and amplification) often occur in human cancers and have been observed in cultured human cancer cell lines [34,35]. At present, however, there is no successful method for freely manipulating ploidy in cultured cells. Although the editing efficiency was comparable among GBM cell lines tested with various numbers of *PDGFRA* ploidy, a detailed investigation of the effect of polyploidy or copy number variation on editing efficiency would be required for further studies. Alternatively, gene variant studies can be examined in monoploid cells, such as HAP-1 cells, because genome-edited monoploid cells may clearly provide a digital phenotype and could clarify the effects of specific gene variants [36,37,38]. However, monoploid or haploidized cancer cell lines do not accurately reproduce the nature of specific cancer cells [39]. Although the establishment of genome-modified cells originating from normal human cell lines would be ideal for carcinogenesis studies, an efficient genome-editing strategy for cancer cell lines of interest should also be established.

In conclusion, genome editing using the Cas9 RNP complex delivered by electroporation into cultured human GBM cell lines is the most effective method for introducing a *PDGFRA* variant. Thus, the Cas9 RNP complex was the first choice for single-nucleotide editing in GBM cells. Efficient editing of genes of interest in human GBM cell lines would enhance our understanding of the carcinogenesis and pathogenesis of GBM. For other human cell lines, an optimal gene editing strategy with varied applications in cells, target genes, and gene alteration types, should be established for further studies on various types of human cancers.

## 4. Materials and Methods

### 4.1. Cell Culture

Information on seven GBM cell lines (U-251 MG, T98G, A172, NMC-G1, KNS-42, SF126, and YKG-1) and the other cell lines used in this study are listed in Appendix A. Cells were maintained at 37 °C in 95% air and 5% CO_2_.

### 4.2. Fluorescence In Situ Hybridization (FISH)

Section 4 μm thick were cut from formalin-fixed, paraffin-embedded (FFPE) cell blocks for FISH analysis. The sections were subsequently immersed in 0.2 N HCl for 20 min, in distilled water for 3 min, and then in 2× saline-sodium citrate buffer (SSC). The glass slides on which the FFPE sections were deposited were microwaved in a pressure jar, followed by protease I digestion (Abbott Laboratories, Des Plaines, IL, USA) for 45 min. The slides were washed twice with 2× SSC, air-dried, fixed with phosphate-buffered neutral 10% formalin for 10 min, and dehydrated with ethanol. Bacterial artificial chromosome clone RP11-231C18 and CEP4 probes (Empire Genomics, Buffalo, NY, USA) were used for *PDGFRA* gene amplification analysis. The sections were then transferred to ThermoBrite (Abbott), which was programmed to perform denaturation at 85 °C for 1 min, after which they were hybridized at 37 °C for 16 h. The slides were air-dried in the dark and counterstained with 4,6-diamidino-phenyl-indole. Images were captured using a fluorescence microscope (BX51; Olympus, Tokyo, Japan), and the red and green fluorescence signals in each section were counted in 30 cells.

### 4.3. Western Blotting

Cells were washed with phosphate-buffered saline (PBS) and precipitated with 10% trichloroacetic acid on ice for 30 min. The precipitates were washed with cold PBS and dissolved in cold lysis buffer (7 M urea, 2 M thiourea, 3% CHAPS, and 1% Triton X-100). The lysates were fractionated by SDS-PAGE and transferred onto polyvinylidene difluoride (PVDF) membranes. The membranes were blocked with 5% nonfat dry milk in Tris-buffered saline containing 0.1% Tween 20 and incubated with the relevant primary antibodies diluted in Can Get Signal solution 1 (TOYOBO, Osaka, Japan). Subsequently, the membranes were incubated with IRDye 680RD donkey anti-rabbit IgG and IRDye 800CW donkey anti-mouse IgG (LI-COR, Lincoln, NE, USA) as secondary antibodies for Western blotting. Fluorescence detection was performed using Odyssey CLx (LI-COR). For chemiluminescent Western blotting, the membranes were incubated with HRP-linked anti-rabbit IgG antibody (#7074; Cell Signaling Technology, Danvers, MA, USA) or HRP-linked anti-mouse IgG antibody (#7076; Cell Signaling Technology), and protein expression was detected using the SuperSignal West Pico chemiluminescent substrate (Thermo Fisher Scientific, Waltham, MA, USA). Protein detection was monitored using a CS Analyzer 3.0 (ATTO, Tokyo, Japan). The following primary antibodies were used: Rabbit anti-PDGFRα monoclonal antibody (#5241; Cell Signaling Technology), rabbit anti-PDGFRβ monoclonal antibody (#3169; Cell Signaling Technology), mouse anti-Cas9 monoclonal antibody (#14697; Cell Signaling Technology), and rabbit anti-β-actin antibody (#4970; Cell Signaling Technology).

### 4.4. Quantitative Reverse Transcription-Polymerase Chain Reaction Analysis

Cells were seeded at 1 × 10^5^ cells/well in 24-well plates (Thermo Fisher Scientific). After the 2-day subculture, total RNA was extracted using the RNeasy kit (QIAGEN, Hilden, Germany) and was converted into cDNA using a ReverTra Ace qPCR RT Master Mix (TOYOBO). The cDNA was amplified by LightCycler 480 (Roche Diagnostics, Basel, Switzerland) using THUNDERBIRD Probe qPCR Mix (TOYOBO). Each sample was analyzed in triplicate in separate wells for the target and reference (*18S* rRNA) genes using the PrimeTime qPCR Probe Assays (Integrated DNA Technologies, Coralville, IA, USA). The average of three threshold cycle values for the target and reference genes was calculated and analyzed using the comparative Ct method.

### 4.5. Plasmid Construction

A plasmid that simultaneously expressed the Cas9 nuclease and the mNeonGreen fluorescent protein (pX330AGmt-1×2) was constructed using the following method. mNeonGreen cDNA was amplified from the mNeonGreen-2A-mTurquoise2 vector (#98885; Addgene, Watertown, MA, USA) and assembled into an all-in-one CRISPR/Cas9 vector (pX330A-1×2; #58766; Addgene) using NEBuilder (New England Biolabs, Ipswich, MA, USA). The BbsI restriction site in the mNeonGreen fragment was mutated (c. 459G>A, p.K153=). The sequence of the sgRNA was “attgttggccaaaatagtcc”, which was designed using CRISPRdirect [40]. The oligonucleotides for sgRNA were annealed and cloned into the pX330Agmt-1×2 vector as reported by Ran [33]. The fragments of cytomegalovirus (CMV) and elongation factor 1 alpha (EF-1α) promoter were amplified from pAcGFP1-c1 (Takara Bio, Shiga, Japan) and PB-UniSAM (#99886; Addgene), respectively, and CMV and EF-1α promoter-driven Cas9 vectors were constructed using NEBuilder (New England Biolabs). Primers used in this study are listed in Appendix A. pAcGFP1-c1 was used as a GFP expression plasmid.

### 4.6. Lipofection

Cells were seeded at 1 × 10^5^ cells/well in 24-well plates (Thermo Fisher Scientific), and the following day, the cells were transfected with Cas9 (500 ng) or GFP (250 ng) expression plasmid using 1.5 μL Lipofectamine 3000 and 1 μL P3000 reagent (Thermo Fisher Scientific) per well according to the manufacturer’s instructions. After the 3-day culture, the cells were harvested for each experiment. For a promoter comparison experiment, 500 ng of CBh, CMV, and EF-1α-driven Cas9 expression plasmids were transfected using the method described above.

### 4.7. Electroporation

The cells were trypsinized and suspended in the Opti-MEM I reduced-serum medium (Thermo Fisher Scientific) at a concentration of 1–2 × 10^7^ cells/mL. The plasmids or RNP complexes were added to 80 μL of the cell suspension, and the mixture was transferred to 2 mm cuvettes (Nepa Gene, Chiba, Japan). For RNP complex electroporation, an electroporation enhancer (1.2 μM, Integrated DNA Technologies) was also added to the cell suspension. Electroporation was conducted using an NEPA21 electroporator (Nepa Gene) with 2× poring pulses and 5× transfer pulses (voltage: 20 V; length: 50 ms; interval: 50 ms; polarity ±). The poring pulse was set according to the cells described in Appendix A. Cas9 RNP complex formation was performed using the following procedure. An equimolar amount of crRNA and fluorescence-labeled tracrRNA (FAM-labeled: Fasmac, Kanagawa, Japan; ATTO488-labeled: Integrated DNA Technologies) were hybridized for 5 min at 95 °C to form sgRNA. Subsequently, sgRNA (1.2 μM) and Cas9 nuclease (1 μM; Integrated DNA Technologies) were mixed with Opti-MEM to form ribonucleoprotein complexes, which were incubated for 30 min at 25 °C. For single-nucleotide substitution experiments, 1 μM of single-strand oligodeoxynucleotides (ssODNs; Integrated DNA Technologies) was mixed with the cell suspension.

### 4.8. Flow Cytometry Analysis and Cell Sorting

The cells were trypsinized and suspended in Hanks’ balanced salt solution containing 1 μg/mL 7-aminoactinomycin D (Nacalai Tesque, Kyoto, Japan). mNeonGreen-positive cells were analyzed using an Attune Nxt flow cytometer (Thermo Fisher Scientific). Images were captured using an Axio Observer fluorescence microscope (ZEISS, Oberkochen, Germany). For cell sorting, fluorescence-positive cells were sorted after gating for doublet and dead cell exclusions using a cell sorter (SH-800S; SONY, Tokyo, Japan) 24 h (Cas9 RNP complex) or 72 h (plasmid) after electroporation. Cells were harvested for genomic DNA extraction after 48 h.

### 4.9. T7 Endonuclease I (T7E1) Assay

Genomic DNA was extracted using the Wizard SV Genomic DNA Purification System (Promega, Madison, WI, USA). The DNA fragment surrounding the target site was amplified using the primers listed in Appendix A. PCR products were denatured in NEBuffer 2 (New England Biolabs) at 95 °C for 5 min and then annealed at a ramp rate of −2 °C/s (95–85 °C) or −0.1 °C/s (85–25 °C). The annealed PCR products were incubated with T7 endonuclease I (T7E1; New England Biolabs) at 37 °C for 15 min, after which they were analyzed by electrophoresis on a 2% agarose gel. Densitometry analysis was performed using a CS Analyzer 3.0 (ATTO). Gene modification was calculated using the following formula: Percentage of gene modification = 100 × [1 − (1 − fraction cleaved)^1/2^].

### 4.10. NGS Analysis

Genomic DNA was extracted using the Wizard SV Genomic DNA Purification System (Promega) and sequenced using a MiSeq sequencer (Illumina, San Diego, CA, USA). Briefly, the sequencing sites were amplified from the genomic DNA using the KOD One PCR Master Mix (TOYOBO) and the primers listed in Appendix A. DNA libraries were prepared from PCR products using a GenNext NGS Library Prep Kit (TOYOBO) and TruSeq DNA Single indexes (Illumina). DNA libraries were purified using solid-phase paramagnetic beads (AMPure XP; Beckman Coulter, Brea, CA, USA). DNA concentration was measured using fluorometric quantification (Qubit; Thermo Fisher Scientific). The alignment of amplicon sequences to a reference sequence and the quantification of the editing efficiency was performed using CRISPResso2 software [41].

### 4.11. Statistical Analysis

The data were plotted and analyzed using R (version 4.1.0) and ggplot2 (version 3.3.3). Three or more replicates were performed for all experiments, and all data are presented as the means ± standard error. Statistical significance was determined using an unpaired one-tailed Student’s t-test, and the results were considered statistically significant at *p* < 0.05.

## Figures and Tables

**Figure 1 ijms-24-00500-f001:**
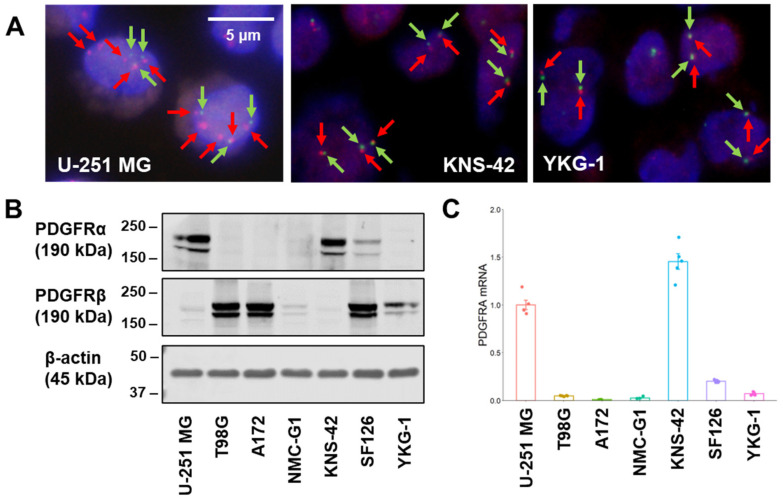
Fluorescence in situ hybridization analysis, protein, and messenger RNA expression in GBM cell lines. (**A**) Representative images of FISH analysis. A probe mix for *PDGFRA* (red arrows, BAC clone RP11-231C18) and CEP4 (green arrows) was employed. Scale bar: 5 μm. (**B**) PDGFRα and PDGFRβ expression levels of GBM cell lines in Western blot analysis. The representative images are shown for experiments that were performed in triplicate. Beta-actin expression is indicated as the loading control. (**C**) *PDGFRA* mRNA expression level was measured by a quantitative reverse transcription-polymerase chain reaction in GBM cell lines. The expression levels were monitored using comparative Ct method with a reference gene expression of *18S* rRNA.

**Figure 2 ijms-24-00500-f002:**
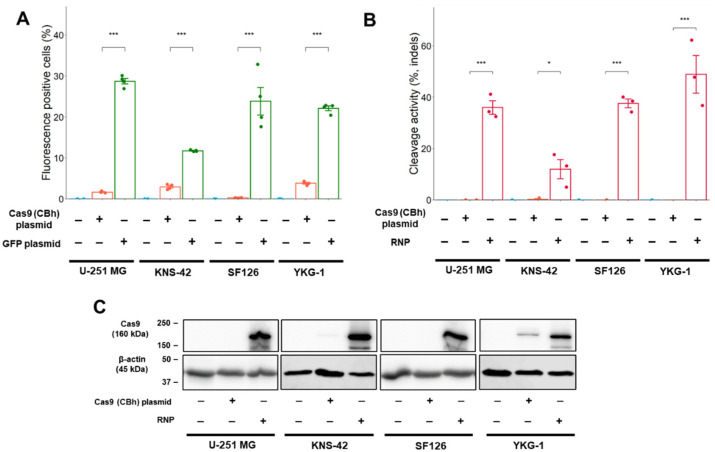
Transfection efficiency and cleavage activity in GBM cell lines. (**A**) Transfection efficiency after plasmid lipofection in GBM cells. The efficiency is indicated by the fluorescence intensity of mNeonGreen and GFP. (**B**) Cleavage activity in GBM cells. The cleavage activity was monitored by T7E1 assay and was determined using densitometric analysis of electrophoresis. (**C**) Cas9 expression in Western blot analysis. The expression was determined 24 h (Cas9 RNP complex) or 48 h (plasmid) after transfection. Numeric data represent means ± standard error (*n* = 4). * *p* < 0.05 and *** *p* < 0.001 vs. Cas9 (CBh) plasmid.

**Figure 3 ijms-24-00500-f003:**
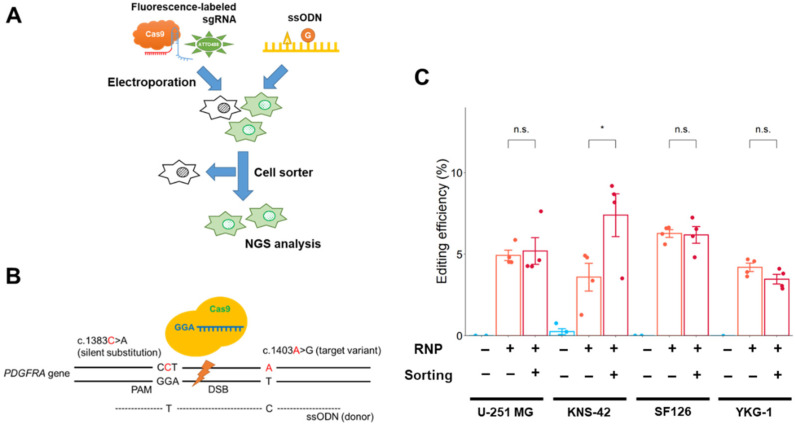
Single-nucleotide editing using the Cas9 RNP complex in GBM cells. (**A**) Schematic workflow of single-nucleotide editing. Cells were sorted by fluorescence-labeled tracrRNA as an indicator 24 h after electroporation. The NGS analysis was conducted after an additional 48 h. (**B**) Schematic representation of the editing region of *PDGFRA* and donor template design. The target nucleotide and silent substitution (for prevention of re-editing) are represented by “c. 1403A>G” and “c. 1383C>A,” respectively. (**C**) Effect of cell sorting on editing efficiency in GBM cells. Data represent the means ± standard error (*n* = 4). * *p* < 0.05 vs. sorting (-). n.s.: not significant.

**Figure 4 ijms-24-00500-f004:**
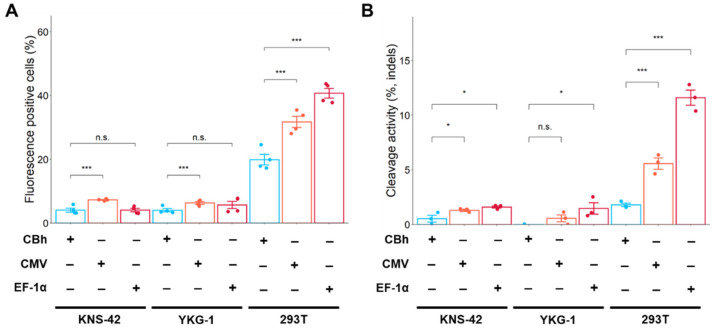
Effects of promoter exchange on fluorescence-positive cells (estimation of Cas9 expression) and cleavage activity in GBM cells and HEK293T cells. (**A**) Fluorescence (mNeonGreen)-positive cells were measured by flow cytometry. (**B**) Cleavage activity was assessed by T7E1 assay. Data represent the means ± standard error (*n* = 4 for flow cytometric analysis, *n* = 3 for T7E1 assay). * *p* < 0.05 and *** *p* < 0.001 vs. CBh. n.s.: not significant.

**Figure 5 ijms-24-00500-f005:**
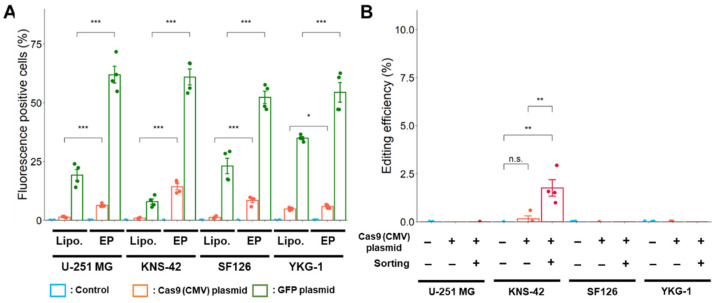
Improvement of transfection and editing efficiencies by electroporation. (**A**) Transfection efficiency after lipofection and electroporation is indicated as the fluorescence intensity of mNeonGreen and GFP as analyzed using a flow cytometer. Lipo.: Lipofection, EP: Electroporation. * *p* < 0.05 and *** *p* < 0.001 vs. lipofection. (**B**) Effect of cell sorting on the editing efficiency with Cas9 expression plasmid by electroporation. Data represent the means ± standard error (*n* = 4). ** *p* < 0.01 vs. sorting (−) or control.

**Figure 6 ijms-24-00500-f006:**
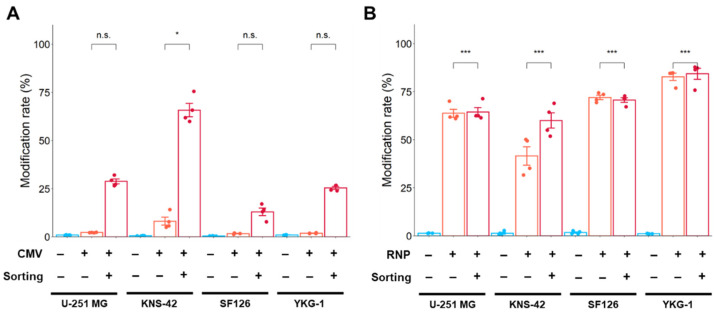
Gene modification rate using Cas9 expression plasmid (**A**) and Cas9 RNP complex (**B**) by electroporation. The modification rate was analyzed by NGS. Modification rate (%) is calculated by the following formula: Modification rate (%) = (number of reads with modification (insertion, deletion, and substitution)/number of total reads) × 100. Data represent the means ± standard error (*n* = 4). * *p* < 0.05 and *** *p* < 0.001 vs. sorting (−). n.s.: not significant.

**Table 1 ijms-24-00500-t001:** FISH analysis of the *PDGFRA* gene in GBM cell lines.

Cell Lines	*PDGFRA*/CEP4		
	Signal Ratio	*PDGFRA* (Red)	CEP4 (Green)
U-251 MG	2.09 ± 0.05	6.00 ± 0.13	2.90 ± 0.06
T98G	1.00 ± 0	3.67 ± 0.14	3.67 ± 0.14
A172	1.00 ± 0	2.34 ± 0.09	2.34 ± 0.09
NMC-G1	1.00 ± 0	2.30 ± 0.09	2.30 ± 0.09
KNS-42	1.00 ± 0	2.20 ± 0.07	2.20 ± 0.07
SF126	1.00 ± 0	2.00 ± 0	2.00 ± 0
YKG-1	1.00 ± 0	2.00 ± 0	2.00 ± 0

CEP4: Chromosome enumeration probe for chromosome 4. *PDGFRA* and CEP4 signals were calculated by counting the number of signals in 30 cells. The CEP4 signal was used to determine the chromosome copy number (ploidy status). The signal ratio was calculated using the following formula: Signal ratio = (*PDGFRA* signal)/(CEP4 signal). Data represent the means ± standard error.

## Data Availability

The data presented in this study are available on request from the corresponding author.

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
