# Peer review of "Genome Editing Using Cas9 Ribonucleoprotein Is Effective for Introducing PDGFRA Variant in Cultured Human Glioblastoma Cell Lines"

_ijms, 2022, doi:10.3390/ijms24010500_

Round 1
Reviewer 1 Report (New Reviewer)
In the manuscript, ‘Genome editing using Cas9 ribonucleoprotein is effective for introducing PDGFRA variant in cultured human glioblastoma cell lines ’ authors presented the efficacy of Cas9 RNP over plasmid expression. They compared the efficiency across several GBM cell lines.
Comments.
1. Number of gene copies is not related to gene expression across the GBM cell line. Could you add a bit of strong evidence such as quantitative RT-PCR on genomic and cDNA? Can you add an image of FISH on a GBM cell with PDGFR/CEP4 signal = 1?
2. Authors designed a plasmid with Cas9 and neon Green. Could you add a schematic figure of this plasmid?
3. Page 2, lines 66-67, can you explain the statement ‘PDGFRA/chromosome enumeration probe for chromosome 4 (CEP4) in the GBM cell lines was 1.00’?
4. In table 1, the signal ratio in most of the GBM cells is 1±0. Can you explain the reason for no variation among replicates?
5. As the authors mentioned around 30 percent electroporation efficiency and post-cell-sorting, we expect all sorted cells must have Cas9 RNP/gRNA. Why do sorted cells not have better editing outcomes? Can authors explain this?
6. In figure 5, we can see very good electroporation efficiency, but no cell lines except KNS-42 show editing. What is the reason behind it?
7. In figure 6, why do authors compare RNP with CMV plasmid, while EF1a showed better efficiency?
8. There are several transduction-based approaches that are efficient. What are the advantages of the approach presented in this manuscript transduction-based approaches?
9. Manuscript lacks the information on off-target effects.
Author Response
We thank the reviewer 1 for your understanding to our study and for the excellent comment aiming to improve the manuscript. We did some additional experiments and modifications according to the reviewers’ suggestions. Please find below our point-by-point responses to the suggestions. The revised parts are shown by “green/red letters” in the revised version.
Point 1: Number of gene copies is not related to gene expression across the GBM cell line. Could you add a bit of strong evidence such as quantitative RT-PCR on genomic and cDNA? Can you add an image of FISH on a GBM cell with PDGFR/CEP4 signal = 1?
Point 2: Authors designed a plasmid with Cas9 and neon Green. Could you add a schematic figure of this plasmid?
Responses: We have revised the manuscripts according to the Reviewer’s suggestions. Please find the additional Figure 1A for a new FISH image, Figure 1C for a quantitative RT-PCR data of PDGFRA mRNA expression, and Supplementary Figure S1A for a vector map. Corresponding parts of the Results, Figure Legend, and Materials and Methods were also revised as below.
Lines 77-79
The expression levels of PDGFR protein well corresponded to those of PDGFRA messenger RNA in the GBM cell lines (Figure 1C).
Lines 94-101
Figure 1. Fluorescence in situ hybridization analysis, protein and messenger RNA expression in GBM cell lines. (A) Representative images of FISH analysis. A probe mix for PDGFRA (red arrows, BAC clone RP11-231C18) and CEP4 (green arrows) was employed. Scale bar: 5 μm. (B) PDGFRα and PDGFRβ expression levels of GBM cell lines in western blot analysis. The representative images are shown for experiments that were performed in triplicate. Beta-actin expression is indicated as the loading control. (C) PDGFRA mRNA expression level was measured by a quantitative reverse transcription-polymerase chain reaction in GBM cell lines. The expression levels were monitored using comparative Ct method with a reference gene expression of 18S rRNA.
Lines 327-336
4.4. Quantitative reverse transcription-polymerase chain reaction analysis
Cells were seeded at 1 × 105 cells/well in 24-well plates (Thermo Fisher Scientific). After 2-days subculture, total RNA was extracted using RNeasy kit (QIAGEN, Hilden, Germany) and was converted into cDNA using a ReverTra Ace qPCR RT Master Mix (TOYOBO). The cDNA was amplified by LightCycler 480 (Roche Diagnostics, Basel, Switzerland) using THUNDERBIRD Probe qPCR Mix (TOYOBO). Each sample was analyzed in triplicate in separate wells for the target and reference (18S rRNA) genes using the PrimeTime qPCR Probe Assays (Integrated DNA Technologies, Coralville, IA). The average of three threshold cycle values for the target and reference genes was calculated and analyzed using the comparative Ct method.
Lines 414-415
Supplementary Figure S1: Vector map of Cas9 expression plasmid (pX330AGmt-1×2) and fluorescent images captured 72 h after lipofection in GBM cell lines;
Point 3: Page 2, lines 66-67, can you explain the statement ‘PDGFRA/chromosome enumeration probe for chromosome 4 (CEP4) in the GBM cell lines was 1.00’?
Response: It was a confusing statement in the manuscript. We revised that part as following.
Lines 67-70
The signal ratio of PDGFRA/chromosome enumeration probe for chromosome 4 (CEP4) in the GBM cell lines was 1.00, except for U-251 MG cells, in which the PDGFRA/CEP4 signal ratio was more than 2.0 (Figure 1A and Table 1).
Point 4: In table 1, the signal ratio in most of the GBM cells is 1±0. Can you explain the reason for no variation among replicates?
Response: Although chromosomal polyploidy was often observed in human cancers, the chromosome 4 was nearly diploid in most of GBM cells used in this study. The reason remains unclear from the results of this study.
Point 5: As the authors mentioned around 30 percent electroporation efficiency and post-cell-sorting, we expect all sorted cells must have Cas9 RNP/gRNA. Why do sorted cells not have better editing outcomes? Can authors explain this?
Response: It would be very unknown phenomenon that we cannot explain now. Further study would be necessary to clarify this. However, at least, we can show the result that Cas9 RNP transfection would be a better gene modification rate even without cell sorting.
Point 6: In figure 5, we can see very good electroporation efficiency, but no cell lines except KNS-42 show editing. What is the reason behind it?
Response: We do not clearly explain why it happens in this study. A transfection efficiency was often inconsistent with the editing efficiency, as shown in Figures 5A and 5B, which might be resulted from the different characteristics among the GBM cell used in this study. We added a sentence as blow.
Lines 186-189
At present, we cannot explain the reason why the discrepancy is observed in transfection and editing efficiency, although it is easy to say that it would be resulted from the different characteristics of the GBM cell lined used in this study.
Point 7: In figure 6, why do authors compare RNP with CMV plasmid, while EF1a showed better efficiency?
Response: The transfection efficiency and cleavage activity were improved with the EF-1α-driven Cas9 expression plasmid (Figures 4A and 4B) in HEK293T cells. However, in KNS-42 and YKG-1 cells, no marked differences in transfection efficiency and cleavage activity were noted between EF-1α- and CMV-driven Cas9 expression plasmid transfection. Therefore, we used the CMV-driven Cas9 expression plasmid for further study in GBM cell lines but not in HEK293T cells.
Point 8: There are several transduction-based approaches that are efficient. What are the advantages of the approach presented in this manuscript transduction-based approaches?
Response: A convenient method using an electroporation-mediated transfer of Cas9 RNP complex would be an advantage of the editing strategy presented here, in which viral-mediated transfection and cell sorting are not always necessary for a successful genome editing. We revised the discussion as blow.
Lines 227-231
In our laboratory, the optimal method for single nucleotide editing of PDGFRA in cultured human GBM cell lines is to use the Cas9 RNP complex, ssODN as donor DNA, and electroporation for transfection, which would be much convenient than the method using viral vector-mediated Cas9 transfer and cell sorting.
Point 9: Manuscript lacks the information on off-target effects.
Response: Kim and colleague reported that a genome editing with a Cas9 RNP complex resulted in reduced off-target mutations compared with plasmid-based genome editing (Kim, S.; Kim, D.; Cho, S.W.; Kim, J.; Kim, J. Highly efficient RNA-guided genome editing in human cells via delivery of purified Cas9 ribonucleoproteins. Genome Res. 2014, 24, 1012-1019; DOI:10.1101/gr.171322.113.). We briefly mentioned a comment for the reduction of off-target mutations with a new reference (ref. no. 24) in the discussion.
Lines 231-233
In this experimental condition, an off-target effect using Cas9 RNP complex is expected to be lower than that using Cas9 expression plasmid [24].
Reviewer 2 Report (Previous Reviewer 3)
The resubmitted version of the manuscript has taken into account my suggestions and is substantially improved. The results are more clearly presented and the conclusions are supported by the provided data.
minor concern
Some grammatical and typographical errors remain throughout the text. Please consider revising. For example:
-line 23: irrespective of cell sorting
-line 120 : 24 h (Cas9 RNP complex) or 48 h
-line 173: using instead of uisng
Author Response
We really appreciate for the reviewer’s comments, which have helped us to improve our paper.
The errors (minor concern) have been corrected in according to the reviewer's comment.
Reviewer 3 Report (Previous Reviewer 1)
I recommended only minor modifications to the previous version/submission of this manuscript. Revisiting my comments, I see that the authors have incorporated what I requested into this new version of the manuscript.
Author Response
We really appreciate for the reviewer’s comments, which have helped us to improve our paper.
Round 2
Reviewer 1 Report (New Reviewer)
I appreciate the efforts the authors made. After extensive revision, the manuscript looks appropriate for consideration.
This manuscript is a resubmission of an earlier submission. The following is a list of the peer review reports and author responses from that submission.
Round 1
Reviewer 1 Report
In this study, Hamada et al. showed that Cas9 ribonucleoprotein can be used for gene-editing in cultured glioblastoma cell lines, specifically through the introduction of a PDGFRA variant in cultured cells. The study appears to have been well conducted, it has scientific significance, and I have only a few minor suggestions:
- Title: Please, put "PDGFRA" in italics (it is a gene name).
- "Mutation" and (gene) "variant" have different meanings in genetics. The authors should review the use of these terms throughout the article as there is confusion with both. I strongly suggest using only the term "variant" since c.1403A>G is a gene variant (of note, the use of "mutation" for gene variants with pathogenic significance in the field of medical genetics is quite questionable since in classical genetics the term "mutation" has an alternative and evolutionary meaning).
Author Response
We would like to thank Reviewer 1 for the helpful and constructive comments on our study. We revised the manuscript following the detailed feedback provided by the Reviewer. We hope that the Reviewer will find our responses below properly address all the raised concerns.
Point 1: Title: put "PDGFRA" in italics
Response 1: We included this modification per the suggestion of the Reviewer.
Point 2: Using only the term "variant"
Response 2: We agree with the Reviewer and incorporated this suggestion throughout the revised manuscript.
Reviewer 2 Report
In the current manuscript Hamada et al., explore the utility of CRISPR-Cas9 system for genome editing in Glioblastoma lines and have presented conditions to optimize genome editing for single nucleotide substitution (PDGFR alpha). They employ CRISPR CAS9 to introduce the mutation and compare it with standard lipofectamine based transfection method. Although some aspects of the optimizations for genome editing may be useful in the current form, the authors are encouraged to consider the following points to improve the manuscript.
The results of Fig 1 – in situ hybridization is confounding. Overall, the authors claim all of them had high copy number of PDGFRalpha but they claim that in contrast 2 had higher ‘gene amplifications”. But technically all of Glioblastoma lines had gene amplification. Then they state that KNS-42 had a “near-normal” copy number and had higher protein expression hence chosen for the study. Fig 1 A is not clear, they should have arrows pointing towards positivity and increase the magnification to appreciate the positivity.
The legend for Table 1 is vague. More explanation is necessary. In Table 1 the signal ratio is 1 for all cell lines except U251-MG when normalised to CEP4. Author claims diploidy and triploidy based on FISH data but the table does not concur with the claims.
The author did not attempt to use AAV which is very efficient in introducing gene edits when combined with CRISPR RNP method. They also did not test lentivirus and other transfecting agents such as PEI.
Fig 3A – not sure what the authors display here as it is very small to understand what they show. They should re-do this to show a magnified panel.
Author Response
We would like to thank Reviewer 2 for the helpful and constructive comments on our study. We revised the manuscript following the detailed feedback provided by the Reviewer. We hope that the Reviewer will find our responses below properly address all the raised concerns.
Point 1: The results of Fig 1 –in situ hybridization is confounding. Fig 1 A is not clear, they should have arrows pointing towards positivity and increase the magnification to appreciate the positivity.
Response 1: We agree with the Reviewer and have incorporated this suggestion in Figure 1A.
Point 2: The legend for Table 1 is vague. More explanation is necessary.
Response 2: Following this comment, we modified the legend of Table 1. The related results (lines 68 to 72) and the legend of Table 1 (lines 92 to 95) are described in more detail.
Point 3: The author did not attempt to use AAV which is very efficient in introducing gene edits when combined with CRISPR RNP method. They also did not test lentivirus and other transfecting agents such as PEI.
Response 3: We agree with the comment of the Reviewer and consider that viral transduction and other transfection reagents (PEI) should be tested in future studies. We included this aspect in the Discussion (lines 242 to 248).
Point 4: Fig 3A – not sure what the authors display here as it is very small to understand what they show. They should re-do this to show a magnified panel.
Response 4: We agree with the Reviewer and incorporated this suggestion in Figure 3A.
Reviewer 3 Report
The Article Titled "Genome editing with Cas9 ribonucleoprotein is effective for introducing the PDGFRA mutation in cultured human glioblastoma cell lines" authored by Taiji Hamada and colleagues, provides evidence on the single nucleotide editing of human glioblastoma cell lines. I have the following Comments and Suggestions for Authors: 1) First, the article is purely technical, presenting a methodology that works in a GBM cell line. As regards the efficiency of the presented assay, the data suggest that KNS-42 was selected because of the optimal expresssion of CRISPR-cas9 components, in contrast to the other cell lines. Why the authors did not try to edit the other cell lines? 2) The manuscript lacks a finding that would warrant the significance of this assay. The aim was to create an experimental tool or to be used as a therapeutic intervention? 3) the manuscript need substantial improvement. to be more specific: -The introduction section feels more like a discussion. The specific information from the bibliography are missing. - There are errors throughout the manuscript and the reading feels like a first draft. For example, in table 1 the text from the MDPI template "This is a table. Tables should be placed in the main text near to the first time they are cited" (lines 82-83) -the discussion section lacks a direction and a specific aim.
Author Response
We would like to thank Reviewer 3 for the helpful and constructive comments on our paper. We revised the manuscript following the detailed feedback provided by the Reviewer. We hope that the Reviewer will find our responses below properly address all the raised concerns.
Point 1: Why the authors did not try to edit the other cell lines?
Response 1: KNS-42 cells, with a nearly normal PDGFRA copy number and higher PDGFRα expression, were selected for gene editing in this study. As the Reviewer indicated, we are also very interested in the genome editing of other GBM cell lines. The comparison among the GBM cell lines for genome editing adaptation would be performed. We mentioned these aspects in the Discussion (lines 224 to 226).
Point 2: The aim was to create an experimental tool or to be used as a therapeutic intervention?
Response 2: We incorporated the comments of the Reviewer into the Introduction (line 49).
Point 3: the manuscript need substantial improvement. to be more specific:
-The introduction section feels more like a discussion.
-There are errors throughout the manuscript and the reading feels like a first draft. For example, in table 1 the text from the MDPI template "This is a table. Tables should be placed in the main text near to the first time they are cited" (lines 82-83)
-the discussion section lacks a direction and a specific aim.
Response 3: We revised the Introduction (lines 51 to 60) and Discussion (lines 209 to 212) sections according to the comments of the Reviewer. We apologize for the typos in the legends of Table 1. These have been corrected.
Round 2
Reviewer 2 Report
I appreciate the author's responses. The major issues raised have been addressed.
Reviewer 3 Report
The results presented remain preliminary. The responses in my comments do not compensate for the lack of experimental data required (Comment 1). The aim of the study should be more clearly addressed in the end of introduction section (Comment 2). The manuscript needs substantial editing to more adequately address my previous concerns (Comment 3). I would happily review a future version of this manuscript that contains all the required data and revisions.